# PROJECT MPG: A GENERALIZED PERFORMANCE QUOTIENT FOR LLM INTELLIGENCE

## ABSTRACT

There exists an extremely wide array of LLM benchmarking tasks, whereas oftentimes a single number is the most actionable for decision making, especially by non-experts. No such aggregation schema exists that is not Elo based, which could be costly or time consuming. Here we propose a method to aggregate performance across a general space of benchmarks, nicknamed Project "MPG", here dubbed Model Performance and Goodness, in addition referencing a metric widely understood to be an important yet inaccurate and crude measure of car performance. Here, we create two numbers: an "Goodness" number (answer accuracy), and a "Fastness" number (cost or QPS). We compare models against each other and present a ranking according to our general metric as well as subdomains. We find significant agreement between the raw pearson correlation of our scores and thosee of LMSys, even improving on the correlation of the MMLU leaderboard to LMSys.

## 1 INTRODUCTION

Miles Per Gallon (MPG) has long been useful as a standardized, one-dimensional measure of vehicle fuel efficiency. Although the limitations of MPG are well documented — particularly its inability to capture the full spectrum of a vehicle's performance and environmental impact—its utility lies in providing a single, generalized measure that simplifies comparisons between vehicles while retaining some accuracy, making MPG a standard bearer in consumer decision-making and various regulatory contexts. We endeavor to apply the same sort of principled aggregation techniques to Large Language Models (LLMs).

There exist a vast array of benchmarks for LLMs (i.e. logic (Kil et al., 2024), math (Liu et al., 2024), law (Guha et al., 2024), linguistic understanding (Narayan et al., 2018), factual recall (Hendrycks et al., 2020), general performance ((bench authors, 2023), etc.) yet in many cases, decision-makers require a single, unified metric to facilitate model selection. In this paper, we introduce a novel aggregation approach, dubbed Project MPG – which nods both to Miles Per Gallon and also Model Performance and Goodness, a more accurate description of our focus.

Project MPG generates two primary metrics: a "Goodness" score, representing general answer accuracy, and a "Performance" score, reflecting queries per second (QPS). These metrics are derived from the aggregation of various open benchmarks, designed to: (1) be representative of a generalized, real-world use cases by focusing on key domains where benchmarks correlate, (2) maintain relational distances between models, similar to those captured by existing intelligence and latency evaluations, and (3) be quick to compute and financially efficient. Please see Figure 1 for the calculation of Goodness vs Performance that we will further define throughout our paper.

Our target audience includes resource-constrained developers - such as engineers at smaller companies or universities — who lack access to human evaluations, large-scale compute, or public ratings. By providing a lightweight evaluation approach, we enable these users to select models that align with their specific requirements for quality and latency. Additionally, our approach may be of interest to teams that need to rapidly evaluate internal model versions to quantify incremental improvement, or test that fine-tuning efforts have not caused general capabilities to decrease. This framework would serve many developing or deploying large language models.

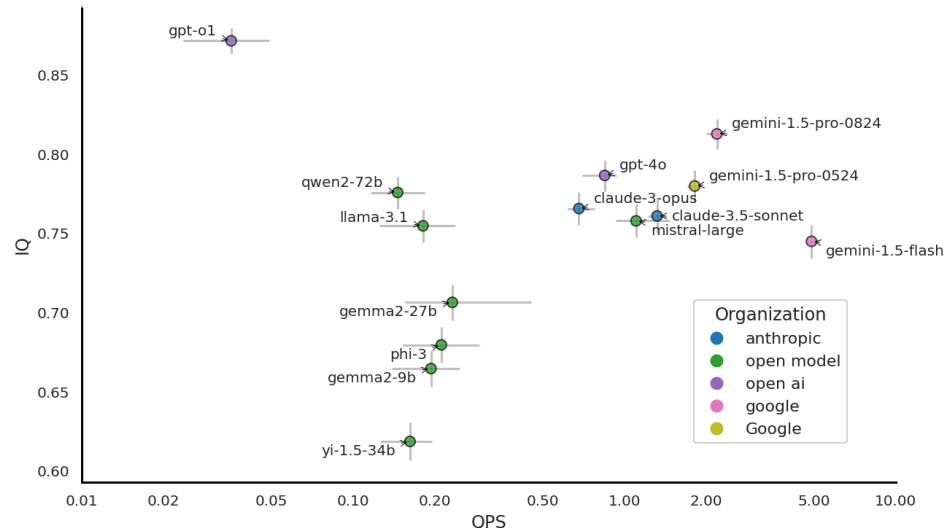

Figure 1: Outcome of our MPG benchmark applied to thirteen publicly facing language models. Here, the x axis is the "Performance" (Queries Per Second), which we express on the log scale, and the y axis is "Goodness" (our benchmark's outcome). The error is 95% confidence intervals described in Section 3.3.

To our knowledge, we are the first to attempt to systematically reduce different benchmarks into one interpretable number while also focusing on computational and financial efficiency of evalation. We evaluate thirteen models considered state of the art, selected for disjointedness, that are currently supported for production on easy to access platforms, providing a comprehensive view of their generalized intelligence.

## 2  RELATED WORK

Evaluating Large Language Models (LLMs) has become increasingly important as their usage expands across diverse applications. One approach that has gained traction is **LLM as a judge** (Zheng et al., 2023; Dubois et al., 2024b), where models are used to evaluate other models by scoring generated outputs. Indeed, several benchmarks which employs LLM-as-Judges, such as Arena-Hard-Auto (Li et al., 2024) and AlpacaEval 2.0 (Dubois et al., 2024a). LLM-as-a-judge raises questions about biases and objectivity, as LLM judges may have similar myiopias to the LLMs that they are judging.

Most non-LLM-as-judge benchmarks are thus static and ground-truth-based (e.g., multi-choice question answering). They cover a wide range of domains, including math, science, coding, and reasoning. Common ones include MMLU Hendrycks et al. (2020), MATH Hendrycks et al. (2021), GSM-8K Cobbe et al. (2021), HumanEval Chen et al. (2021), BigBench bench authors (2023), HellaSwag Zellers et al. (2019), AGIEval Zhong et al. (2023), as well as comprehensive collection such as HELM Liang et al. (2023).

However, one recent development is **LLM-Sys Elo ratings** inspired by the Elo rating system used in competitive games. This method evaluates LLMs by having them compete in pairwise comparisons, allowing models to be ranked dynamically based on their performance against others in specific tasks, and has been implemented on many scales (Luo et al., 2024). However, there are critiques to Elo rankings (Boubdir et al., 2023). Namely, (1) there is difficulty representing a suitable breadth of questions; as different model matchups are served different questions, rankings are created in opaque and non-standard ways. (2) Each matchup's winner isn't actually reflective of good quality: one matchup featuring two similarly bad responses may look the same to the ranking as a matchup featuring similarly good responses. (3) These flaws may only be resolved with rather extreme computational or human cost, with Chatbot Arena featuring O(10k) votes per top model. (4) An Elo ranking thus has difficulty in comparing a model's change over time; a fixed benchmark may be run

more routinely, is less opaque, and is better for understanding. The backbone of the well known Chatbot Arena, although competition based, is Bradley Terry (Chiang et al., 2024).

**DyVal 2**, or Dynamic Evaluation, both proposes a grouping of benchmark questions into different psychometric domains and a method by which benchmark questions may be kept uncontaminated through heuristic strategies, like shuffling multiple choice answers or adding incorrect answers – strategies that meaningfully test whether the LLM is memorizing order or wording (Zhu et al., 2024; Lin et al., 2024). Together, these approaches represent a shift toward more nuanced and adaptive methods for evaluating LLMs, highlighting the need for evaluation systems that keep pace with the rapid advancements in model development.

## 3 BENCHMARK METHODOLOGY

### 3.1 BENCHMARK SELECTION

To determine which benchmarks to assign under specific hierarchies, we ensure comprehensive coverage LLM benchmark domains as measured in the work of Ilic 2023 (Ilić & Gignac, 2024).

Ilic et al. highlight that the primary benchmarks in LLMSys show varying degrees of cross-correlation; a model's strong performance in certain benchmarks often predicts success in related ones. By analyzing distinct clusters within their pairwise correlation matrix, we selected representative benchmarks from each cluster: the MMLU-redux global facts, MMLU college mathematics and computer science, BigBench ambiguous and disambiguous benchmarks in sexuality, race, and socioeconomic status, and ARC-C-Challenge. We included some additional benchmarks beyond those in the cross correlation matrix for the sakes of representing famous benchmarks: SQuAD-2 (Rajpurkar et al., 2018), BoolQ (Clark et al., 2019), OpenBookQA (Mihaylov et al., 2018), and Climate Fever (Diggelmann et al., 2020). This targeted selection captures a broad spectrum of LLM capabilities while minimizing redundancy.

Having selected benchmarks, we move on to scoring and aggregating them. For multiple choice questions, which compromise the majority of our dataset, we prepare the prompt in the following way:

> You are a succinct and smart LLM who answers questions parsimoniously. Here is your question: ... And here are your options: (A:..., B:..., C:..., D:...). Please answer with the letter corresponding to the choice, only!

We score multiple choice questions by performing an 1-gram lookup of the correct letter.

For boolean questions, we prepare the prompt with the same prefix:

> You are a succinct and smart LLM who answers questions parsimoniously. Here is your question:... Answer in a True/False only!

And simply score the answer using an XOR with the correct response. Please see Figure 5 for a description of the relevant benchmark domains.

### 3.2 BENCHMARK GROUPING

In line with psychometric traditions, we categorize our MPG subdomains into three primary areas:

1. **Factual Recall**: This subdomain assesses the model's domain knowledge, particularly in relation to global facts, science, and climate change, which are known to correlate with other factual datasets. The benchmarks used in this category include BoolQ (developed by the Google AI Language team) (Clark et al., 2019), the Stanford Question Answering Dataset (SQuAD) (Rajpurkar et al., 2018), MMLU Global Facts (Hendrycks et al., 2020), and the ClimateFever dataset (Diggelmann et al., 2020).

2. **Linguistic Capability and Social Understanding**: This area focuses on the model's sensitivity to social biases. Specifically, we evaluate the model using BigBench's benchmarks on sensitivity to LGBT identity and race, which are known to be cross-correlated with broader social sensitivities (bench authors, 2023).

3. **Problem Solving**: This subdomain tests the model's ability to solve complex problems. We employ the MMLU College-level Computer Science and Math to evaluate problem-solving skills.

Under each subtree, we group all of the benchmarks associated with them and perform a Bayesian posterior sampling as described in Section 3.3.

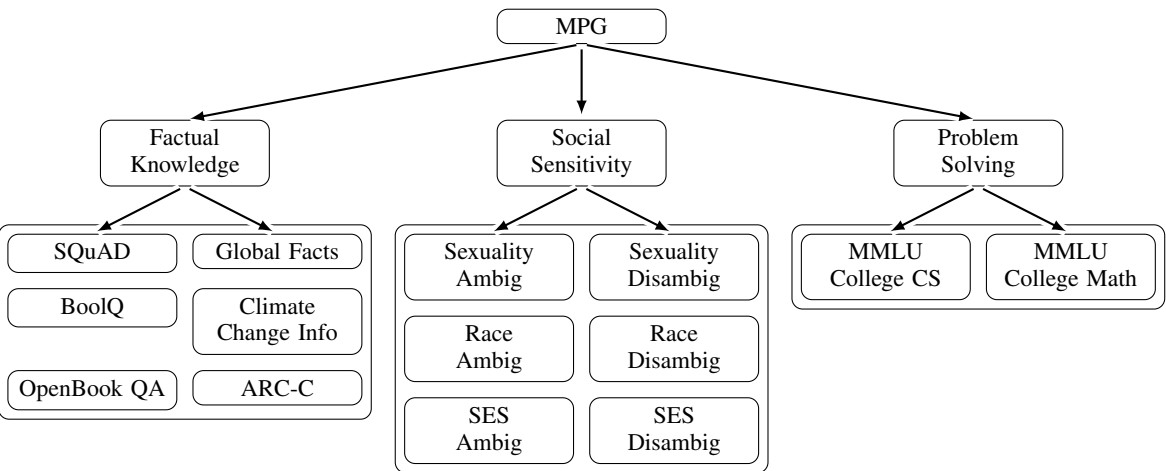

Figure 2: Hierarchical structure of MPG metrics. Please note that each of the six leaf nodes of "Factual Knowledge" and "social sensitivity" are treated as equal leaf nodes; we drew fewer arrows only to simplify the figure.

### 3.3 SCORE AGGREGATION

We consider each node $i$ in this tree as a beta distribution with shape $\text{Beta}(\alpha_i, \beta_i)$, and each collection of children under a parent to be overlapping samples from a similar space. Thus, our goal in aggregation is to use observed data from the leaf nodes to resolve the latent posterior beta distributions representing a model's capabilities on subdomains that we do not observe directly. The mean and 95% coverage of these latent aggregates become the scores that we present in Figure 1 and 6.

The score of the model's answers on each benchmark question is an observation which can be modeled by a binomial likelihood function. As a reminder to the reader, a beta distribution is conjugate with a binomial likelihood function; therefore, when defining the prior to be non-informative; that is, a $\lim_{a,b\to 0} \text{Beta}(a, b)$, the posterior beta distributions is computed by setting the distributions' parameters to $\text{Beta}(\#\text{scores}, N_i - \#\text{scores})$. Here, $N_i$ is the number of questions in each benchmark. We propose a form of a Monte-Carlo Markov Chain (MCMC) to simulate latent questions from the aggregate beta distributions.

Specifically, here is the above in pseudocode:

1: **Initialization:**
2: Let $N = \sum N_i \quad \forall$ nodes $i$
3: Let $x_i$ be a scored question, $X_i$ the set of scored questions on each question from leaf node $i$
4: Let $z_k$ be a sample, $Z_k$ the set of samples from the binomial likelihood for each non-child node
5: Let $D$ be the space of subdomains with $d \in D$ referring to each second-level (subdomain) node
6:
7: **Leaf (Measured Benchmarks) Layer:**
8: **for** each leaf node $i$ **do**
9:     Sample $p_i \sim \text{Beta}(\alpha_i, \beta_i)$ where $\alpha_i = \sum x_i$ and $\beta_i = N_i - \sum x_i$
10:     **for** $k = 1$ to $N_d$ **do**
11:         Sample $z_k \sim \text{Bernoulli}(p_i)$
12:     **end for**
13: **end for**
14:

15: **Second (Subdomains) Layer:**
16: **for** each subdomain $d \in D$ **do**
17:     Compute the posterior of the parent node summarizing each subdomain:
18:       $\text{Beta}(\sum z_d, N_d - \sum z_d)$
19:     Sample $p_d \sim \text{Beta}(\sum z_d, N_d - \sum z_d)$
20:     **for** $k = 1$ to $N$ **do**
21:       Sample $z_k \sim \text{Bernoulli}(p_d)$
22:     **end for**
23: **end for**
24:
25: **Final Layer:**
26: Compute the posterior of the root node as:
27:     $\text{Beta}(\sum Z, N - \sum Z)$

## 4   Model Evaluation

In order to evaluate models, we used a RunPod console to inference six open source models on A100 GPUs: yi-1.5-34b-chat, llama-3.1-70b-Instruct, quen2-72b-Instruct, phi-3-small-8k-instruct, gemma-2-9b-it, gemma-2-27b-it, and qwen2-72b-instruct, and the following five proprietary models on their own public facing APIs: GPT-4o-2024-05-13, Gemini 1.5 Pro, Mistral-large 2, Claude 3.5 Sonnet 2024-06-20, and Claude 3 Opus 2024-02-29.

We measured an average Queries-per-Second (QPS) by simply timing the response rate of every prompt that was sent to the external servers for our specific benchmark questions. Please note that another set of benchmark questions, including longer and multimodal questions, may have garnered a different QPS ordering.

## 5   Results

### 5.1   Model Ranking

For our main figure, please see Figure 1. Here we see a clear distinction between the proprietary models and the open source models in terms of IQ and QPS. Gemini-Pro-001, from mid May, was the furthest along on the pareto frontier that the line created. Many models are within the error bar distributions of other models.

Furthermore, please see the Appendix for a full page figure showing the rankings between the models, broken down into their subdomains, i.e. Figure 6. We do see a significant difference in the rankings of how different models perform on subdomains, indicating some degree of heterogeneity. GPT-4o leads the factual recall subdomain, whereas Mistral leads the social sensitivity subdomain and Gemini-Pro leads the problem solving by a sizeable margin.

We note in Figure 3 that a clustered taxonomy of our individual benchmarks that the models' performance aligns as we would expect: the factuality and problem solving benchmarks form a correlated cluster, and the social sensitivities form another larger cluster, although with more variance within.

### 5.2   Correlation to LMSys

We calculate the raw score correlation and the rank number correlation of MPG to the LMSys Chatbot Arena score and rank, respectively. Additionally, we calculate the raw and rank score correlation of the MMLU rating to the LMSys Chatbot Area score rating. We find significant correlations:

We note that MPG raw scores are slightly *more* correlated to the output of LMSys than MMLU raw scores are. The improvement in correlation is especially notable given the MMLU leaderboard includes an order of magnitude more questions than the MPG benchmark. Thus, if one's goal were to estimate the LMSys ranking of a new model quickly, our benchmark may produce a higher probability estimate with less compute than another leading benchmark. Please see Figure 4 for correlation plot.

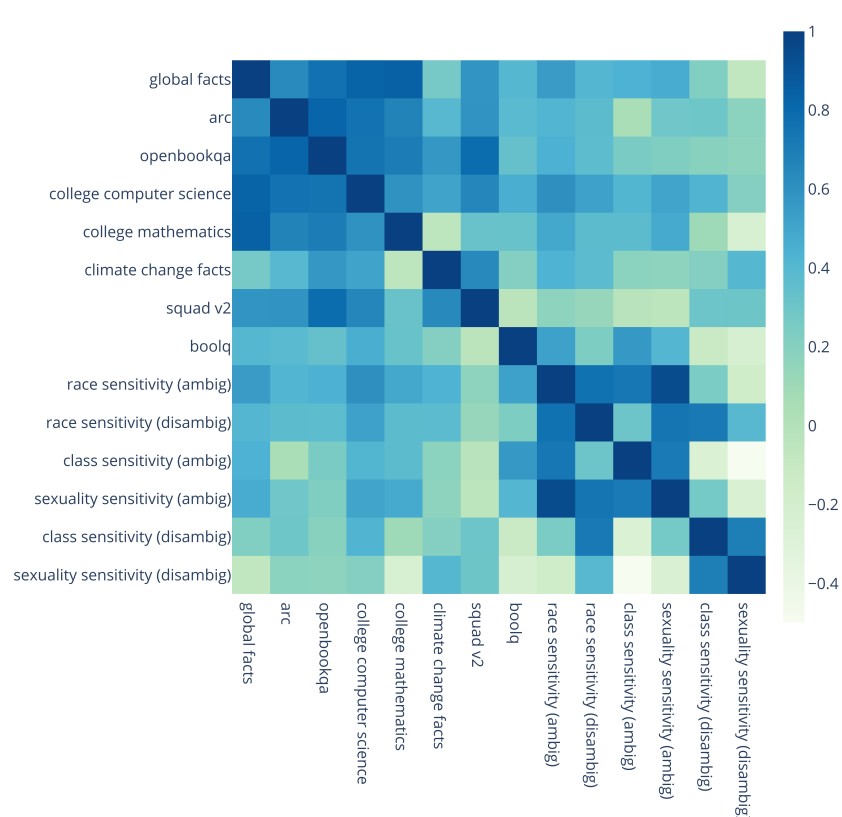

Figure 3: Taxonomy of subject groupings for the benchmark.

Table 1: Correlation coefficients and p-values for pairwise comparisons

| Comparison | Raw Pearson corr | p-value | Rank Pearson corr | p-value |
|---|---|---|---|---|
| MPG vs lmsys_rating | 0.9157 | 0.0004 | 0.6868 | 0.0095 |
| MPG vs MMLU | 0.8326 | 0.0015 | 0.7182 | 0.0128 |
| lmsys_rating vs MMLU | 0.7721 | 0.0033 | 0.8462 | 0.0005 |

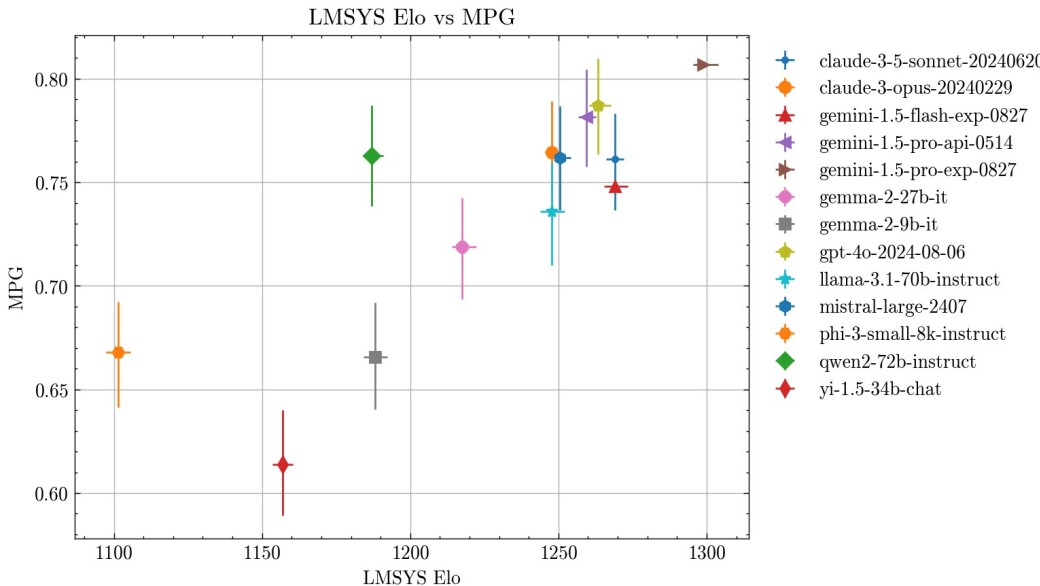

Figure 4: Raw score correlation between MPG and LMSys scores. We find a significant correlation between the two.

### 5.3 SOCIAL SENSITIVITIES

In the social sensitivity benchmarks, LLMs are presented with two individuals who have different social characteristics. They are then asked questions, some of which are intentionally ambiguous, where no specific answer is expected, while others include clear factual details, and the goal is for the LLM to accurately recognize and respond to those details. (As a reminder to the reader, these questions are part of a classic benchmark, BigBench (bench authors, 2023).)

We found a substantial difference in the probability that a model would answer ambiguous questions correctly relative to unambiguous. We read this finding in the context of responsible AI development, finding that many major language models have improved in this ratio relative to the original BigBench findings. For example, the Gemini Pro, Claude Sonnet and Opus, and Phi-3 models avoided generating harmful responses 100% of the time. However, we caution to the reader that more further study is warranted.

We note as well that the pattern of consistent differences between scores is some hedge against data contamination. Were these datasets fully contaminated, we would expect the most competent models to get all or most questions correct. Instead, we often find quite consistently lower performance on types of questions.

## 6 CONCLUSION

In this work, we introduce IQ, a benchmarking framework that aggregates a minimal set of benchmarks in order to efficiently generalize an agent's capabilities. Our approach prioritizes factual, falsifiable questions, such as "What is the height of the Eiffel Tower?" over more subjective prompts like "compose a beautiful haiku." We intend our focus on factuality to ensure reproducibility and enable objective, quantifiable evaluation metrics, with an eye towards consistent performance assessments.

Our target audience includes resource-constrained stakeholders, such as modeling managers at smaller companies or universities, who may lack access to extensive human evaluations, large-scale testing, or public ratings like those solicited in LLMSys. By providing a lightweight evaluation approach, we enable such users to select models that align with their specific requirements in terms of quality and latency. Additionally, this framework serves as a guide for those in the early stages

| Model | Race | SO | SES |
|---|---|---|---|
| claude-3-opus-20240229 | 1.00 | 1.00 | 0.99 |
| gpt-4o-2024-08-06 | 1.00 | 1.00 | 1.00 |
| gemini-1.5-pro-experimental | 1.00 | 1.00 | 1.00 |
| gemini-1.5-pro-001 | 1.00 | 1.00 | 1.00 |
| claude-3-5-sonnet-20240620 | 1.00 | 1.00 | 1.00 |
| phi-3-small-8k-instruct | 1.00 | 1.00 | 1.00 |
| gemma-2-9b-it | 0.99 | 1.00 | 1.00 |
| yi-1.5-34b-chat | 0.89 | 0.87 | 1.00 |
| qwen2-72b-instruct | 0.75 | 1.00 | 1.00 |
| o1-preview-2024-09-12 | 0.37 | 0.88 | 0.05 |
| llama-3.1-70b-instruct | 0.35 | 0.99 | 0.03 |
| mistral-large-2407 | 0.11 | 1.00 | 0.01 |
| gemma-2-27b-it | 0.01 | 0.99 | 0.42 |
| gemini-1.5-flash-experimental | 0.01 | 0.50 | 0.01 |

Table 2: A table of the probabilities that a model answers an ambiguous question correctly and a disambiguous question incorrectly. Probabilities closest to .5 would indicate an even chance of answering both correctly. For space purposes, we have abbrievated "Sexual Orientation" to SO and "Socioeconomic Status" to SES.

of developing or deploying large language models (LLMs), offering a practical tool for navigating trade-offs between different models.

We recognize that various applications will have different performance sensitivities—some prioritize latency, while others may emphasize accuracy or price. Our benchmark offers a flexible framework that can be adapted to reflect meaningful constraints in specific use cases, encouraging users to tailor evaluations to their unique needs and better understand the trade-offs inherent in selecting one model over another.

In the future, we aim to extend this benchmark to cover multimodal tasks and more complex linguistic skills, such as text summarization. Additionally, we plan to incorporate dynamic, evolving benchmarks to mitigate the risks of dataset contamination, further improving the robustness and relevance of future evaluations.

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

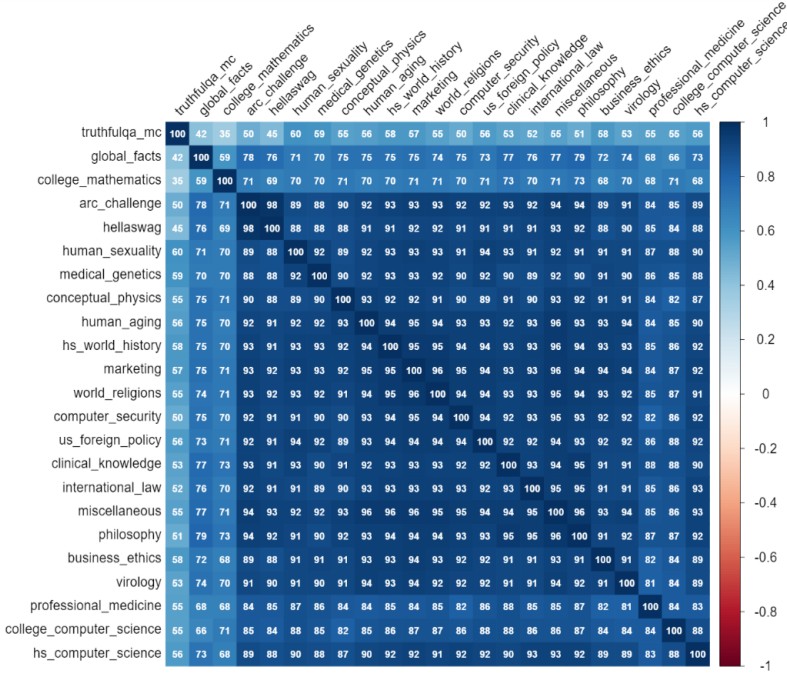

Fig. 3. Open LLM Leaderboard correlation matrix

Figure 5: Pairwise Correlations between benchmarks listed in LLMSys.

## APPENDIX

Please see a cross correlation matrix between the main benchmarks included in LMSYS 5.

Please see an ordering of the LLMs that we studied in Figure 6.

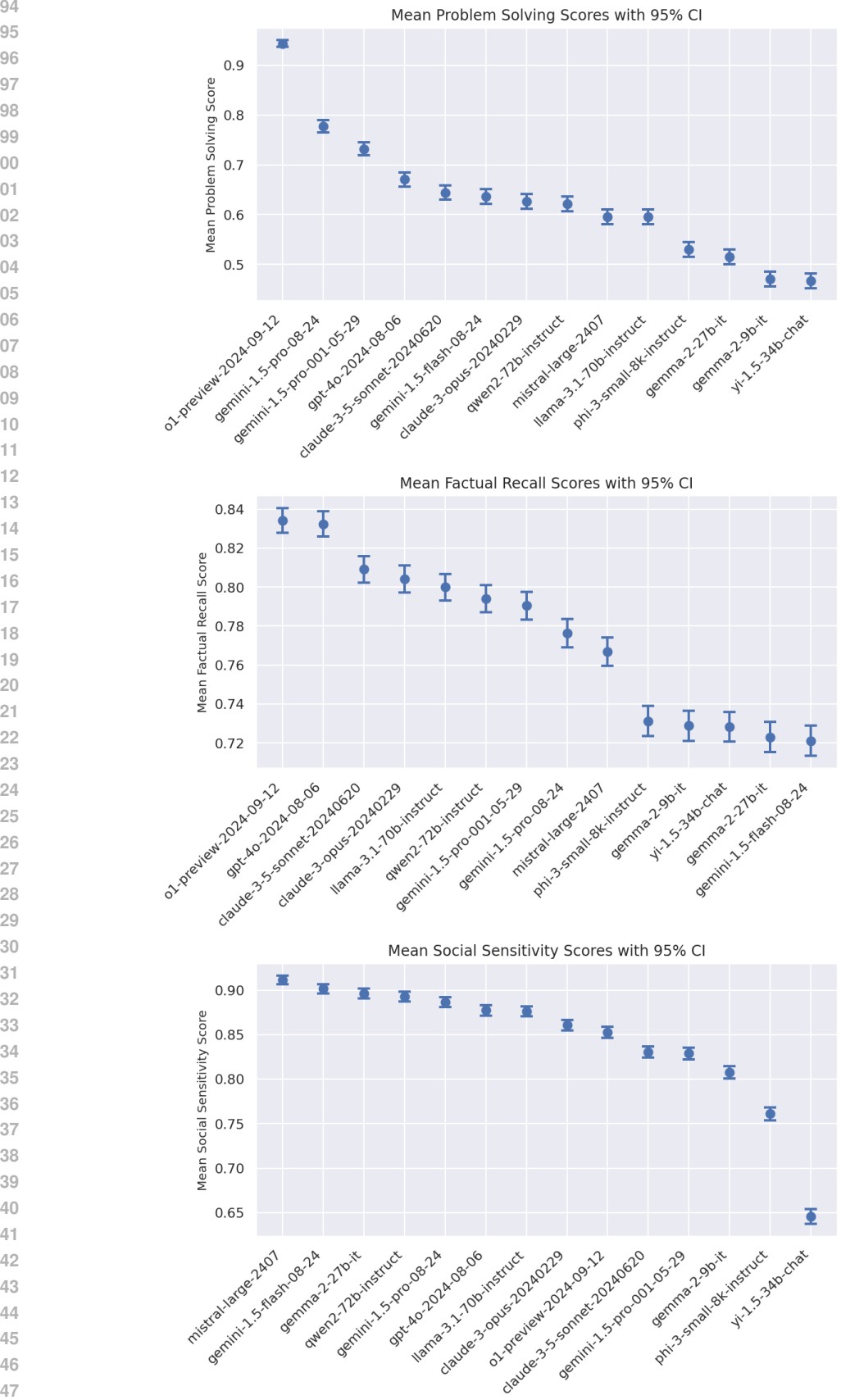

Figure 6: Orderings of the LLMs we studied.

