# OpenReview forum: "Project MPG: towards a generalized performance quotient for LLM intelligence"
_ICLR.cc/2025/Conference — Submitted to ICLR 2025_

### Official Review · Reviewer_WCkM · 2024-10-16

**Soundness:** 2
**Presentation:** 2
**Contribution:** 1
**Rating:** 1
**Confidence:** 4

**Summary:**

The paper introduces a new performance measurement for LLMs and tests it on some of the well-known LLMs. The measurement is two dimensional: one is for the response accuracy and the other is for the efficiency (QPS).

**Strengths:**

This paper's topic is of interest since the community is suffering from the lack of a golden standard to measure an LLM's true performance.

**Weaknesses:**

This paper appears to lack a solid foundation in terms of both theoretical motivation and empirical rigor. The idea seems to be introduced without substantial grounding, and the experimental methodology appears somewhat ad-hoc, merely applying the approach to several well-known LLMs without providing accompanying code, open-source datasets, or a thorough explanation of the measurement's validity. Moreover, the vague correlation presented in Figure 3, which is the main result, leaves the effectiveness of the measurement uncertain. Based on these limitations, I do not recommend accepting this paper.

**Questions:**

Please refer to the weakness part.

---

### Official Review · Reviewer_piQz · 2024-10-21

**Soundness:** 2
**Presentation:** 1
**Contribution:** 2
**Rating:** 1
**Confidence:** 4

**Summary:**

This paper introduces Project "MPG", an approach to measuring the capabilities and efficiency of Large Language Models. The main contribution of the work is to formalise two metrics: "Goodness" and "Fastness". Fastness is defined as the average response rate for an LLM when queried. Goodness, as far as I can tell, is a latent capability inferred from the distribution of performances across a series of benchmarks that purport to measure different capabilities. It looks like the authors use a hierarchical Bayesian model to infer this latent capability quotient. First, they define a hierarchy of nested capabilities, where the terminal nodes are candidate benchmarks (e.g., MMLU College Computer Science). In their case, they group benchmarks into three intermediary capabilities (Factual Knowledge, Social Sensitivity, Problem Solving), which are then grouped into a single Goodness capability. Second, They fit a Bernoulli to each item from each benchmark, representing the probability of success on that item. They then use that as a likelihood to fit a Beta distributed posterior for the intermediate capability. Then those posteriors are used to infer a Beta distributed posterior of Goodness, giving a single number that they take to denote the accuracy, or intelligence, of the LLM. This methodology allows us to aggregate performances from many different benchmarks with many different measurement schema, for which simply averaging over benchmark performance would lead to biased capability estimates.

The authors put their metrics to use, defining a Pareto frontier between goodness and fastness. This allows researchers using MPG to evaluate models by balancing performance and efficiency. The MPG methodology is generalisable beyond the benchmarks and capabilities that the authors employ here, meaning that there is scope for future extensions that study different capabilities or use more complete test beds.

**Strengths:**

The paper has a number of strengths. First, the paper tackles an important lacuna in LLM research - accuracy is being prioritised over efficiency. The community must move towards measurement frameworks that balance performance against the efficiency of models. A preference for performance incentivises developers to constantly scale up their models and continue to use a vast number of resources. Benchmarks and schema like MPG that also incentivise efficiency can help to kickstart progress towards efficient architectures. Second, the authors cover a number of state-of-the-art LLMs, including closed- and open-source models, which is commendable. Third, they situate their work within the context of a number of other approaches and include comparisons between them and Project MPG.

**Weaknesses:**

The paper contains many weaknesses.

First, the presentation of the paper is poor. There are numerous spelling errors, syntactic disfluencies, and semantic ambiguities throughout. This makes it very difficult to follow the paper and discern the authors' methodology and the novel contributions of the paper.

Second, I find the presentation of the 'Fastness' number as problematic. It is defined as the average time taken to respond to every prompt sent to external servers for benchmark questions. This seems incredibly noisy as a measure because it is contingent on the connection speeds of the authors, the traffic on the various servers being used, and rate limits on the proprietary APIs. I am not sure how reproducible these numbers would be and whether the QPS values taken from RunPod instances of open-source models are commensurable with the QPS values from the different APIs (e.g., Anthropic, OpenAI, Google). The authors are right to remark that the numbers would be different and give different rankings depending on the benchmarks and the questions, but there are so many other variables to consider here too. In any case, I would have liked to see the QPS values being normalised by the number of tokens in the query. I think that would go some way to improving comparability between benchmarks and between models. It is also not clear which kind of averaging is used to compute QPS, but I would recommend the geometric rather than the arithmetic mean since these are rates.

Third, the authors present their latent capability analysis as being in some sense novel. However, there is an extensive literature that proposes methods such as theirs, with added sophistication. In the cognitive sciences, the field of psychometrics has produced a number of latent capability estimation procedures, most notably, Item Response Theory. The proposal in this paper is also closely related to (Bayesian) Structural Equation Models which are used to infer latent abilities and cognitive capacities in social and developmental psychology and computational psychiatry. In recent years, these methodologies have been used and extended in the field of AI Evaluation, including for LLM capability evaluations. The authors should look at the papers listed at the end of this section. In particular, Burden et al. (2023), Kipnis et al. (2024), Polo et al. (2024) and Wang et al. (2023).

Fourth, the capability hierarchy in Figure 2 seems arbitrary. Finding names for the latent constructs is a common problem in latent factor estimation, but usually such hierarchies are accompanied by independent evidence for why certain test sets are deemed to probe certain capabilities and not others. Understandably, the MPG project and methodology extends beyond this exact capability hierarchy, but since it is in the paper, it needs some motivation.

Fifth, the methodology for latent capability estimation is difficult to follow. Is this a hierarchical Bayesian network? Is it a structural equation model with priors? What does it mean when the authors say 'we propose a form of a Monte-Carlo Markov chain (MCMC) to simulate latent questions from the aggregate beta distributions.' It is unclear what latent questions are.

Finally, the paper seems to end by ignoring the QPS measure and focus mostly on the latent capability measure and how well it recovers LMSys scores, etc. Much more time ought to be spent on the idea of a Pareto frontier between the two components of MPG and determining how we might create a balanced metric between these two facets. This is all left unsaid.

References

Burden, J., Voudouris, K., Burnell, R., Rutar, D., Cheke, L., & Hernández-Orallo, J. (2023). Inferring Capabilities from Task Performance with Bayesian Triangulation. arXiv preprint arXiv:2309.11975.
Wang, X., Jiang, L., Hernandez-Orallo, J., Stillwell, D., Sun, L., Luo, F., & Xie, X. (2023). Evaluating general-purpose ai with psychometrics. arXiv preprint arXiv:2310.16379.
Hernández-Orallo, J. (2017). Evaluation in artificial intelligence: from task-oriented to ability-oriented measurement. Artificial Intelligence Review, 48, 397-447.
Hernández-Orallo, J. (2017). The measure of all minds: evaluating natural and artificial intelligence. Cambridge University Press.
Kipnis, A., Voudouris, K., Buschoff, L. M. S., & Schulz, E. (2024). metabench--A Sparse Benchmark to Measure General Ability in Large Language Models. arXiv preprint arXiv:2407.12844.
Polo, F. M., Weber, L., Choshen, L., Sun, Y., Xu, G., & Yurochkin, M. (2024). tinyBenchmarks: evaluating LLMs with fewer examples. arXiv preprint arXiv:2402.14992.
Pellert, M., Lechner, C. M., Wagner, C., Rammstedt, B., & Strohmaier, M. (2023). AI Psychometrics: Using psychometric inventories to obtain psychological profiles of large language models. OSF preprint.
Ye, H., Xie, Y., Ren, Y., Fang, H., Zhang, X., & Song, G. (2024). Measuring Human and AI Values based on Generative Psychometrics with Large Language Models. arXiv preprint arXiv:2409.12106.

**Questions:**

1. How does this work fit into the much larger body of literature on latent capability estimation in the cognitive sciences and AI/machine learning?
2. How do we balance Goodness and Fastness? I like the idea of a Pareto Frontier. This is mentioned once on page 5 but is not further discussed.
3. The presentation ought to be clearer. The conclusion does not appear to match the introduction for instance - the authors state that they 'introduce IQ' there, but at the start they state that their contribution is the MPG metric.
4. The methodology for latent capability estimation is difficult to follow. Is this a hierarchical Bayesian network? Is it a structural equation model with priors? What does it mean when the authors say 'we propose a form of a Monte-Carlo Markov chain (MCMC) to simulate latent questions from the aggregate beta distributions.' It is unclear what latent questions are.

In general, I think that the paper would require considerable work to reach publishable quality. In any case, I do not think that the content (or what I have understood of it) is particularly novel.

---

### Official Review · Reviewer_zwty · 2024-11-02

**Soundness:** 2
**Presentation:** 1
**Contribution:** 2
**Rating:** 1
**Confidence:** 4

**Summary:**

The paper proposes a unified metric, "Model Performance and Goodness" (MPG), to benchmark LLMs based on their "Goodness" (accuracy) and their "Fastness" (queries per second, QPS). The authors aim to provide a cost-effective and computationally efficient metric that could serve users with limited resources, such as smaller companies or university labs. The authors evaluate 13 models and compare MPG's correlation to other benchmarks, such as LMSys and MMLU, demonstrating a fair level of correlation.

**Strengths:**

Clear Motivation: The paper effectively highlights the gap in the benchmarking field, specifically the lack of an accessible, single aggregated metric that balances accuracy and computational efficiency. The intention of this benchmark is generally a good one.
Practical Application: The target audience is well-defined, and the authors justify the metric's utility for smaller organizations and rapid internal testing. If done right, this could be a helpful resource.

**Weaknesses:**

There are actually many studies showing that MPG is not a good measure of gas consumption. See, for example, here:
https://www.science.org/doi/full/10.1126/science.1154983
https://journals.sagepub.com/doi/abs/10.1177/237946151500100109

The introduction already misses out on some related papers, such as:
tinybenchmarks: https://arxiv.org/abs/2402.14992 -> one of the first papers attempting to reduce benchmark size
metabench: https://arxiv.org/abs/2407.12844 -> essentially a very similar effort, measuring abilities via one number but over many more LLMs and --from what I can tell-- more principled?


For the score aggregation, the tree is just assumed (not estimated or derived). It's like imposing structure from the beginning, while this structure could have also been estimated. One can also see later that the factuality and problem solving benchmarks formed a cluster.
"We measured an average Queries-per-Second (QPS) by simply timing the response rate of every
prompt that was sent to the external servers for our specific benchmark questions." -> I think this might cuse all sorts of problems where time might not only reflect how long the query took.

All of page 6 is a heatmap and a table, it really feels like there just isn't enough here for an ICLR paper. I think this would be better as a workshop paper.

In the conclusion section, the project is --all of a sudden-- called IQ. I think this really seems to show that this has been a rushed submission. The first paragraph seems to be from an earlier version of the paper, whereas the second paragraph seems to be an exact repetition of what had been said before.

**Questions:**

Why is the y-axis of Figure 1 presented as IQ? Please refrain from using this concept here. Also, this seems to be a relict from an older version of this paper.
It's unclear what Figure 3 shows. I assume it's Pearson's r?
Some of the rank correlations aren't actually that high. Perhaps the benchmark isn't that good overall?

---

### Official Review · Reviewer_DXqD · 2024-11-04

**Soundness:** 1
**Presentation:** 3
**Contribution:** 1
**Rating:** 3
**Confidence:** 3

**Summary:**

The authors propose a method/metric that resource-constrained users can use to choose an LLM that best serves their purpose. They name it ``MPG'' to underline how one cannot summarize a performance with a single value.

**Strengths:**

- The paper is well-written and easy to follow
- They get good correlations with LMSys (a difficult benchmark to re-produce due to online ranking LLM answers) for their benchmark evalution method, improving upon MMLU (however, see weaknesses as well).

**Weaknesses:**

- If I'm not mistaken, I believe the starting point for the paper and the solution are not consistent. Especially the part where, who will need such a unified score to make a selection for which LLM to use, and why they cannot simply use other benchmarks. If a developer wants to compare different in-house models (that are not available online), can't they simply use offline benchmarks that are strictly related to the metrics they care about? Why would they need a unified score that may include unrelated metrics (such as social sensitivity)?

**Questions:**

- Why is QPS there to begin with? It's subject to change for APIs, and even for open-source models, different quantized models will vary hugely in terms of speed. It's also not something novel, so I don't understand giving any importance to it even in the abstract
- On table 1, MMLU seems to have better Rank Pearson corr value to lmsys_rating, casting a doubt on the usefulness of MPG
- Why can't someone simply evaluate their model (for publicly available models, one can simply use LMSys ratings), on their specified benchmarks and make the choice that way? Why would they need MPG?

---

### Meta-Review · Area_Chair_PoAG · 2024-12-19

**Metareview:**

The reviewers agree that this paper is well motivated and attempts to address an important issue of benchmark aggregation. However, all reviewers unanimously recommend rejection, due to lack of theoretical and empirical soundness, inconsistency in reasoning, missing important related work. Thus, I would not recommend accepting this paper based on its current state. I would encourage the authors take recommendations from the reviewers to further improve the paper.

**Additional Comments On Reviewer Discussion:**

Reviewers unanimously reject the paper. The authors did not respond to reviews.

---

### Decision · Program_Chairs · 2025-01-22

Reject